# Fractionation of Arctic Brown Algae (*Fucus vesiculosus*) Biomass Using 1-Butyl-3-methylimidazolium-Based Ionic Liquids

**DOI:** 10.3390/molecules28227596

**Published:** 2023-11-14

**Authors:** Artyom V. Belesov, Daria A. Lvova, Danil I. Falev, Ilya I. Pikovskoi, Anna V. Faleva, Nikolay V. Ul’yanovskii, Anton V. Ladesov, Dmitry S. Kosyakov

**Affiliations:** Laboratory of Natural Compound Chemistry and Bioanalytics, Core Facility Center ‘Arktika’, Northern (Arctic) Federal University, 163002 Arkhangelsk, Russia; darya.lvova2017@yandex.ru (D.A.L.); i.pikovskoj@narfu.ru (I.I.P.); a.bezumova@narfu.ru (A.V.F.); n.ulyanovsky@narfu.ru (N.V.U.);

**Keywords:** brown algae, *Fucus vesiculosus*, ionic liquids, 1-butil-3-methylimidazolium, biomass fractionation

## Abstract

Arctic brown algae are considered a promising industrial-scale source of bioactive sub-stances as polysaccharides, polyphenols, and low-molecular secondary metabolites. Conventional technologies for their processing are focused mainly on the isolation of polysaccharides and involve the use of hazardous solvents. In the present study a “green” approach to the fractionation of brown algae biomass based on the dissolution in ionic liquids (ILs) with 1-butil-3-methylimidazolium (bmim) cation with further sequential precipitation of polysaccharides and polyphenols with acetone and water, respectively, is proposed. The effects of IL cation nature, temperature, and treatment duration on the dissolution of bladderwrack (*Fucus vesiculosus*), yields of the fractions, and their chemical composition were studied involving FTIR and NMR spectroscopy, as well as size-exclusion chromatography and monosaccharide analysis. It was shown that the use of bmim acetate ensures almost complete dissolution of plant material after 24 h treatment at 150 °C and separate isolation of the polysaccharide mixture (alginates, cellulose, and fucoidan) and polyphenols (phlorotannins) with the yields of ~40 and ~10%, respectively. The near-quantitative extraction of polyphenolic fraction with the weight-average molecular mass of 10–20 kDa can be achieved even under mild conditions (80–100 °C). Efficient isolation of polysaccharides requires harsh conditions. Higher temperatures contribute to an increase in fucoidan content in the polysaccharide fraction.

## 1. Introduction

Marine micro- and macroalgae produce unique biogenic compounds possessing a wide range of biological activities, which makes them a promising source of pharmaceuticals, food supplements, cosmetics, antimicrobial agents, fertilizers, etc. This is especially true for brown algae, which are widespread in the seas of temperate and high latitudes and produce polyphenolic compounds, polysaccharides (fucoidan, alginic acids), and other valuable secondary metabolites in large quantities [1]. Due to harsh environmental conditions at high latitudes and long daylight hours during the growing season, arctic brown algae are capable of producing the largest amounts of bioactive substances and are of greatest interest as a feedstock for industrial-scale processing (biorefining). 

Among the most important constituents of brown algae are polyphenols, which are present both as low molecular weight compounds and, predominantly, as phloroglucinol- based polymers—phlorotannins. The latter group of compounds possesses pronounced antioxidant properties, and has antidiabetic, antiproliferative, anti-HIV, radioprotective, and anti-allergic effects [1,2,3]. The content of polyphenols in the algal biomass varies depending on the brown algae species, age, and growth location, and can reach 20% of the dry weight (d.w.) [4]. The highest values were observed in the brown algae, namely *Fucus vesiculosus* (15–18%) and *Ascophyllum nodosum* (14–15%), growing in the White and Barents Seas [5]. Thus, isolation and subsequent valorization of polyphenols must be an integral component of brown algae biorefining technologies. Despite this, the technologies traditionally used for the industrial processing of macroalgae are aimed primarily at isolating the polysaccharide component and, in some cases, obtaining pigment (chlorophyll and carotenoids) extracts. Conventional polysaccharide extraction techniques typically involve treating the algal material with various solvents such as hot water or acidic or salt solutions at high temperatures for several hours [6] after preliminary removing of lipids with nonpolar extractants, including toxic halogenated hydrocarbons (chloroform, dichloromethane). 

The development of novel approaches to algal biomass processing is hindered by the lack of effective methods for selective extraction of target components from plant raw material and their separation and purification. At the same time, due to growing environmental safety concerns, the developed technologies should rely on the use of “green” processes and solvents. The latter include, first of all, sub- and supercritical water and carbon dioxide, ionic liquids (ILs), and deep eutectic solvents [7,8,9]. Unusual physical and chemical properties of ILs (ionic structure, extremely low vapor pressure, incombustibility, thermostability) and high dissolving power toward various classes of biomolecules, as well as the possibility of efficient regeneration allow considering ILs as most promising media for algal biomass treatment. Moreover, the well-known ability of ILs to completely dissolve plant tissues [10] through breaking inter- and intramolecular bonds in biopolymers (mainly ether bonds between phenolic units) opens prospects for the efficient fractionation of algal biomass into polysaccharide and aromatic components [11,12] as the basis for biorefining technologies.

Dissolution of algal biomass in ionic liquids has been described previously for various algae species [13,14,15,16], including those belonging to brown algae, such as *Sargassum fulbellum*, *Laminaria japonica*, *Undaria pinnatifida*, *Saccharina japonica*. Dialkylimidazolium- and alkylpyridinium-based room temperature ILs (chlorides, tetrafluoroborates, and acetates) were used at temperatures of 100–150 °C and at a treatment duration of up to 6 h. However, the main objective of these studies was the isolation of polysaccharides for subsequent hydrolysis with further biofuel production, as well as lipid extraction with organic solvents without focusing on the obtaining and characterizing polyphenolic fractions. 

In the present study, we propose an approach to the fractionation of brown algae biomass involving its dissolution in dialkylimidazolium ionic liquids with further selective antisolvent precipitation of polysaccharides and polyphenols. 1-Butyl-3-methylimidazolium (bmim) methyl sulfate ([bmim]MeSO_4_), chloride ([bmim]Cl), and acetate ([bmim]OAc), which significantly differs in anion basicity and solvation properties, and have previously proven themselves well in solving problems of wood processing to produce cellulose and lignin [17], were chosen as studied ILs. Bladderwrack (*Fucus vesiculosus*) brown algae species, most widespread and harvested on an industrial scale in the White Sea, was used as an object of research aimed at optimizing conditions for algal biomass fractionation, isolating, and characterizing the resulting polyphenolic and polysaccharide fractions. 

## 2. Results and Discussion

### 2.1. Solubility of Algal Biomass 

The nature of IL anion and temperature of the reaction mixture are the most critical factors determining the completeness and dynamics of dissolving plant biomass. In our experiments, the powdered bladderwrack thallus samples were treated with the three ILs at 80, 100, 120, and 150 °C under constant stirring with the measurements of the solid residue during 24 h (the attained relative standard deviation was 10–15%). The obtained results (Table 1) demonstrate that the substantial dissolution of the plant material (>50%) can be achieved at temperatures ≥ 100 °C. Treatment at 150 °C for 24 h allowed dissolving up to 92% of the bladderwrack biomass when using [bmim]OAc as a solvent. 

The comparison of the three ILs revealed two different patterns of their action depending on the treatment temperature. At higher temperatures (120–150 °C) the pronounced dependence on the IL’s anion nature is observed, and the studied ILs, according to their effectiveness, can be arranged in the following series, corresponding to a decrease in the basicity of the anion [18,19]: [bmim]OAc > [bmim]Cl > [bmim]MeSO_4_. The higher basicity of bmim acetate and chloride is expected to break hydrogen bonds in biopolymers and promotes dissolution of polysaccharides due to specific interactions with acetate and chloride anions. Chemical (covalent) interactions with the IL cation, occurring with the participation of a reactive carbene intermediate [20], also may contribute to the dissolution of biopolymers. The formation of carbene occurs due to the deprotonation of bmim upon interaction with highly basic anions and is promoted at elevated temperatures. 

Treatment at 80 °C enables the dissolution of up to 30% of the algal biomass, and the solubility in different ILs varies as follows: [bmim]OAc ≈ [bmim]MeSO_4_ > [bmim]Cl. This effect may be explained by the ability of the most basic (bmim acetate) and acidic (bmim methyl sulfate) ILs to cleave ether and ester bonds in biopolymers. At this temperature, it is likely that the polyphenolic component is preferentially dissolved in ILs, with the proportion of the polysaccharide component increasing as the duration of treatment increases (mainly for [bmim]OAc and [bmim]MeSO_4_). At a temperature of 100 °C, a transitional system state is observed—the effectiveness of [bmim]Cl increases more significantly than for other ILs and thus the differences in their action are not so pronounced as at higher or lower temperatures. Considering the previous data on the thermostability of the ionic liquids used [20], prolonged heating of the reaction mixture at 120–150 °C cannot be recommended for [bmim]OAc, which is prone to partial decomposition under these conditions. However, samples isolated with [bmim]OAc at 120–150 °C were further used for comparison purposes.

According to the recorded FTIR spectra (Figure 1), the insoluble residue obtained after IL treatment consists of polysaccharides, predominantly alginates. 

This is evidenced by intense absorption bands corresponding to carboxylate anions observed at ~1600 and ~1400 cm^−1^ (asymmetric and symmetric C-O-C stretching, respectively) along with a number of peaks in the area of C-O single-bond stretching (1200–1000 cm^−1^). The absence of signals from aromatic compounds confirms the assumption about the preferential transition of polyphenols into the IL liquid phase. 

### 2.2. Biomass Fractionation 

For the selective isolation of polysaccharide and polyphenol components from the obtained solutions of plant material in ILs, a known approach involving sequential precipitation with different antisolvents was used. As regards the latter: Following the example of wood fractionation [17], acetone and water, which can effectively precipitate polysaccharides and phenolic compounds, respectively, were chosen. The advantages of acetone over some other organic solvents are its rather low environmental toxicity and availability as a bio-based solvent, which allows for considering this substance as a green solvent if properly recycled in the technological process [21,22]. Fractions F1 (insoluble in acetone) and F2 (insoluble in water) obtained after 24 h treatment in accordance with Figure 1 were separated from the liquid phase of the reaction mixture with yields of up to 39% (F1) and 11% (F2) from the initial plant material (Table 2). 

The presented data indicate that all the studied ILs are able to extract almost completely the polyphenolic component already at 80–100 °C. This is evidenced by the yield of the F2 fraction, which corresponds to the polyphenol content of the studied commercially available *Fucus vesiculosus* preparation. In this regard, it’s not surprising that changing the processing temperature has no substantial effect on the yield of the F2 fraction. The insignificant decrease in the content of polyphenols isolated from the [bmim]OAc solution obtained at 150 °C may be caused by thermal degradation of phlorotannins or solvolysis of ether bonds in their structure. 

Increasing the processing temperature had a more pronounced effect on the yield of polysaccharide fraction (F1). Thus, when going from 80 to 150 °C, at least a twofold gain in the content of isolated polysaccharides was attained, which, however does not correspond to the increase in the dissolved biomass percentage (for example, from 30 to 90% in the case of [bmim]OAc). This may be evidence that the polysaccharide component is partially hydrolyzed under the action of ILs in the presence of residual water. The greatest differences were observed for the samples treated with [bmim]MeSO_4_ and [bmim]Cl, which is explained by the presence of strong acid anions and thus occurrence of the acidic medium promoting hydrolysis. 

The chemical composition and properties of the isolated fractions are the most important factors that determine their suitability for practical applications. Since the highest degree of biomass dissolution was achieved after prolonged heating in ILs, the acetone (F1) and water (F2) insoluble fractions obtained after 24 h plant material treatment at various temperatures were used for further characterization by Fourier transform infrared (FTIR) and nuclear magnetic resonance (NMR) spectroscopy, size-exclusion chromatography, and ligand exchange chromatography for monosaccharide determination.

### 2.3. Chemical Composition of Fraction F1

FTIR spectroscopy measurements confirmed the predominance of the polysaccharides in fraction F1 (Figure 2). 

All the investigated samples showed broad absorption bands at 3500–3300 cm (hydrogen bonded O-H stretching) and typical signals of aliphatic C-H stretching of 3000–2800 cm^−1^. The presence of a polysaccharide backbone is confirmed by an intense peak centered at 1025–1010 cm^−1^, which is attributed to C-O-C in glycoside bridge stretching [23]. Due to overlap with other peaks related to the stretching of the carbon-oxygen single bond in C-O-C, C-O-H, and C-O-S (in fucoidan) structures, the intense broad absorption band is observed in this region (1010–1090 cm^−1^) [24,25,26]. Identification of fucoidan in fraction F1 is also based on the combination of specific bands associated with sulfur–oxygen bonds at ~830 (C-O-S stretching) and 1250–1220 (S-O stretching in sulfate esters) [26]. The peak at ~1620 cm^−1^ arises from asymmetric O-C-O stretching in carboxylates (alginic acid). The noticeable peak at ~1750 cm^−1^ (C=O stretching vibrations) may indicate the presence of acetyl moieties [27]. At the same time, the presented FTIR spectra do not contain significant signals of aromatic compounds. This allows for concluding that the fraction F1 comprises polysaccharides (alginic acid, fucoidan), which completely coincides with our expectations.

For more detailed characterization, the polysaccharide fraction was subjected to acid hydrolysis with further monosaccharide determination. The obtained data (Table 3) indicate a decreased proportion of hydrolysable polysaccharides in fraction F1 isolated by using bmim chloride as a solvent. Another peculiarity is an increase in fucose (fucoidan) content with increasing temperature of the IL treatment, which may indicate the preferential isolation of alginic acid. The most selective IL for fucoidan extraction is [bmim]MeSO_4_, which provides the highest (up to 279 mg g^−1^) fucose content in the fraction F1 hydrolysates. The high glucose content is considered evidence of the presence of cellulose in noticeable amounts along with alginic acid and fucoidan.

The fraction F1 obtained after the [bmim]OAc treatment at the highest temperature (150 °C) exhibits a relatively high hydrolysable polysaccharide content (277 mg g^−1^) along with the highest yield from the plant material (39%). It is worth noting that a sharp increase in the fucoidan content is observed when the IL treatment temperature increases from 120 to 150 °C, whereas the yield of the fraction increases insignificantly. This means that high treatment temperatures promote the transition of fucoidan into the solution, while alginic acid is extracted by bmim acetate under mild conditions. 

The predominance of alginic acid (alginates) in the non-hydrolysable part of the fraction F1 was confirmed by FTIR spectra of the residue after acid hydrolysis (Figure 3).

A distinctive feature of alginates are intense absorption bands at ~1610 and ~1450 cm^−1^, which are characteristic of the carboxylate anion and related to asymmetric and symmetric C-O stretching, respectively [27]. Weaker bands at 1700–1710 cm^−1^ may be attributed to the molecular form of carboxylic acids (C=O stretching) and considered additional evidence of the alginic acid presence in the studied sample. Other intense bands in the FTIR spectra presented in Figure 3 are related to vibrations of the O-H and C-O groups typical of polysaccharides and mentioned above in the discussion of the F1 fraction spectra (Figure 2). 

### 2.4. Chemical Composition of Fraction F2

Preparations containing high levels of polyphenolic compounds are of thegreatest interest. FTIR spectra of fraction F2 show the predominance of aromatic components in its composition (Figure 4) responsible for the intense absorption bands at ~1620 and ~1517 cm^−1^ corresponding to C-C stretches in the aromatic ring. Other most intense peaks at 1260–1100 cm^−1^ related to C-O-C stretching allow for attributing them to phlorotannins. Rather intense absorption at ~1450 and 1370 cm^−1^ (asymmetric and symmetric C-H bending in aliphatic structures, respectively) may indicate the presence of methyl or other hydrocarbon moieties as substituents or admixtures originating from other types of lipophilic extractives present in the algal biomass—carotenoids, steroids, lipids, etc. In general, the recorded FTIR spectra correspond well to those published in the literature for algal phlorotannins [28,29]. Considering that the yield of the acetone-soluble fraction is close to the content of polyphenols in the plant material, this gives reason to believe that fraction F2 is almost completely represented by phlorotannins and contains only minor admixture of other lipophilic compounds.

This finding is confirmed by molecular weights and molecular weight distributions of the F2 fractions obtained by size-exclusion chromatography. As can be seen from Table 4, weight-average molecular weight (M_w_) values of all the obtained fraction F2 preparations fall into a rather narrow range of 10–20 kDa. 

The relatively low values of polydispersity indices (PDI) indicate the small proportion of low-molecular compounds and, as a consequence, the high homogeneity of the samples. It should be noted that an increase in the treatment temperature, especially in the range 120–150 °C, leads to an increase in the molecular weights of the isolated phlorotannins. This is explained by the side processes of polyphenol condensation at higher temperatures. The latter can be promoted by acidic medium; thus it is natural that the samples obtained with more acidic bmim chloride and methylsulfate at 150 °C demonstrate higher polymerization degrees when compared to [bmim]OAc. 

For a more detailed characterization of the polyphenolic fractions, their two-dimensional HSQC (heteronuclear single quantum coherence) NMR spectra were recorded and analyzed for the samples obtained after 24 h treatment at 150 °C (Figure 5). 

The residual presence of ILs in the sample results in the most intense signals on the spectrum. However, their chemical shift is located in the non-target region of the spectrum and does not interfere with the identification of target compounds. To determine the possibility of extracting polyphenolic compounds from algae using ILs, it is necessary to analyze the aromatic region of the spectrum at δC/δH 90–140/5.0–7.5 ppm. It is known that the main polyphenolic compounds of algae are phlorotannins, which are characterized by clusters of signals with chemical shifts at 90–100/5.5–6.0 ppm, which is clearly observed in the experimental spectrum. In addition, the spectrum shows clear signals in the aromatic region, which can be correlated with the structures of amino acids such as phenylalanine and tyrosine based on their chemical shifts (δC/δH 110–140/6.5–7.3 ppm). However, identification of all amino acids present in fraction F2 is currently impossible. Further research is underway. It is worth noting that the obtained spectra do not show any signals corresponding to polysaccharides.

## 3. Materials and Methods

### 3.1. Reagents and Materials

1-Butyl-3-methylimidazolium acetate, chloride, and methyl sulfate (BASF quality, >95%) were purchased from Sigma-Aldrich (Steinheim, Germany). Algae biomass fractionation was carried out with the use of “chem. pure” grade acetone (Komponent-Reaktiv, Moscow, Russia) and deionized water obtained with the Milli-Q system (Merk Millipore, Molsheim, France).

Brown alga bladderwrack (*Fucus vesiculosus*) thallus plant material was purchased from the Arkhangelsk Seaweed Plant (Arkhangelsk, Russia) as a dried and powdered commercial product with the following characteristics: alginates—35.4%, other carbohydrates—20.0%, ash—22.1%, polyphenols—10.0%, proteins—7.0%, water—4.0%, fats—1.5%. 

### 3.2. Algal Biomass Fractionation

An accurately weighed 50 mg sample of algal biomass was placed in a 2.5 mL glass vial containing 1 mL of IL. The mixture was subjected to a thermal treatment at 80–150 °C for 2–24 h under constant agitation. After the treatment, the reaction mixture was separated by centrifugation into an insoluble residue fraction (if present) and a solution in IL. A 5-fold excess of acetone was added to the obtained solution, and the resulting precipitate (fraction F1) was separated from the liquid phase by filtration. Then the acetone was removed from the solution under a vacuum on a rotary evaporator, and a dark viscous liquid with a volume of ~1 mL was obtained. The fraction F2 was isolated through precipitation by adding 5-fold excess of water, cooling to 4 °C, and leaving it overnight in nitrogen atmosphere, followed by filtration. Both obtained fractions were dried in a vacuum oven to the constant weight. To enhance the reliability of the obtained results, at least three repetitions of the biomass dissolution and fractionation procedure were performed. 

### 3.3. Size-Exclusion Chromatography

Determination of the molecular weights and molecular-weight distributions of the polyphenolic fractions (F2) was carried out by size-exclusion high-performance liquid chromatography using an LC-20 Prominence HPLC system (Shimadzu, Kyoto, Japan) consisting of an SIL-20A autosampler, an LC-20AD pump, a DGU A3 vacuum degasser, an STO-20A column thermostat, and an SPD-20A spectrophotometric detector. Separation was performed at 50 °C on a Polargel-M chromatographic column (Agilent, Santa Clara, CA, USA), 300 × 7.5 mm. Lithium bromide solution (0.0125 M) in DMF was used as a mobile phase with a flow rate of 1 mL min^−1^. Detection was performed at a wavelength of 275 nm. The system was calibrated with monodisperse polystyrene standards (PSS, Mainz, Germany) in a molecular weight range of 0.35–187 kDa.

### 3.4. Monosaccharide Analysis

Six target monosaccharides (glucose, xylose, galactose, arabinose, mannose, fructose) were determined by high-performance ligand exchange chromatography (HPLEC) with refractometric detection according to the procedure described earlier [30]. A Nexera XR HPLC system (Shimadzu, Kyoto, Japan) which consisted of a DGU-5A vacuum degasser, an LC-20AD chromatographic pump, a SIL-20AC autosampler, a CTO-20AC column thermostat, and an RID-20A refractometric detector was used. The chromatographic separation was carried out at 75 °C on a Rezex RPM-Monosaccharide Pb^+2^ column (Phenomenex, Torrance, CA, USA), 300 × 7.8 mm, using the pure water (flow rate 0.6 mL min^−1^) as a mobile phase. The injection volume was 10 µL. The system control and quantification of the analytes were performed using LabSolution software ver. 5.71 SP1 (Shimadzu, Kyoto, Japan). The HPLC system was calibrated using the aqueous standard solutions of the monosaccharide mixture with concentrations of 10–1000 mg L^−1^. 

The total monosaccharide content was determined after a preliminary two-stage acid hydrolysis of the extracts according to the following procedure. The dry extract sample (10 mg) was placed in a 4 mL conical glass vial, poured with 100 µL of 72% sulfuric acid, and kept at 30 °C for 60 min in a Reacti-Therm reaction system (Thermo Scientific, Waltham, MA, USA) equipped with a heating block and a magnetic stirring module. Then, 2.5 mL of water was added and the reaction mixture was heated to 100 °C, kept for 3 h under continuous stirring, and allowed to cool down at ambient conditions. After neutralizing the acid by adding an excess of BaCO_3_ and centrifugation, the obtained solution was injected to the HPLC system. All assays were performed in triplicate.

### 3.5. FTIR and NMR Spectroscopy

IR spectra were obtained on a Vertex 70 IR Fourier spectrometer (Bruker, Bremen, Germany) equipped with a GladiATR attenuated total reflection (ATR) system (Pike Tech., Madison, WI, USA) with a diamond prism. The spectra were recorded under the following conditions: spectral range 4000–400 cm^−1^, resolution 4 cm^−1^, 128 scans. The resulting IR spectrum was subjected to ATR correction with transformation to absorbance units. The instrument was controlled and the spectra were processed with an OPUS software package ver. 8.2.28 (Bruker, Bremen, Germany).

The NMR spectra were recorded on an AVANCE III NMR spectrometer (Bruker, Ettlingen, Germany) with a working frequency for protons of 600 MHz. The ^1^H-^13^C HSQC (Heteronuclear Single Quantum Correlation) spectra were recorded using the following parameters: temperature—298 K, spectral window width ~13 ppm for F2 and ~200 ppm for F1 with a number of accumulations—1024 × 256, number of scans—8, delay time between pulses (D1)—2.0 s. To register the ^1^H-^13^C HSQC spectra, about 40 mg of sample was dissolved in 0.5 mL of DMSO-d_6_.

## 4. Conclusions

1-Butyl-3-methylimidazolium-based ionic liquids possess high dissolution power towards brown algae biomass and can be used for its fractionation into separate polysaccharide and polyphenolic constituents within the biorefinery concept. The effectiveness of ILs as solvents is determined by the nature of anion and increases in the following series corresponding to an increase in the anion basicity: [bmim]MeSO_4_ < [bmim]Cl < [bmim]OAc. The use of the latter IL ensures dissolution of most bladderwrack (*Fucus vesiculosus*) plant material at temperatures above 120 °C and up to 92% during 24 h treatment at 150 °C. The fractionation strategy involving sequential precipitation of the fractions from IL solution with acetone and water allows for obtainment of the polysaccharide mixture (alginates, cellulose, and fucoidan) and polyphenols (phlorotannins) with the yields of ~40 and ~10%, respectively. Polyphenolic fraction does not contain significant amounts of carbohydrates and can be near-quantitatively extracted from the algal biomass by any of the three studied ILs even under mild conditions (80–100 °C). It contains mainly phlorotannins with a weight-average molecular mass of 10–20 kDa and rather low polydispersity (PDI = 1.6–2.5), as well as an admixture of low-molecular lipophilic substances. Efficient isolation of polysaccharides requires harsh conditions of IL treatment and the use of bmim acetate as a biomass solvent. Higher temperatures facilitate isolation of fucoidan, while alginic acids can be extracted under milder conditions. 

Our conceptual study opens up prospects for the development of new approaches for algal biomass valorization for the eco-sustainable production of various products. In this regard, further research should be focused on the detailed characterization of the obtained fractions, optimization of IL treatment conditions, and the development of effective procedures for separation of individual polysaccharides and biologically active low-molecular-weight secondary metabolites. 

## Data Availability

Data are contained within the article.

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
