# Peer review of "Fractionation of Arctic Brown Algae (Fucus vesiculosus) Biomass Using 1-Butyl-3-methylimidazolium-Based Ionic Liquids"

_molecules, 2023, doi:10.3390/molecules28227596_

Round 1

Reviewer 1 Report

Comments and Suggestions for Authors

A fine paper describing the specific algae dissolution and fractionation using BMIM ionic liquids.The results are well explained and the methodology makes sense. A major question remains as to why these ILs were selected and especially their stability at elevated temperatures, which was not verified.

Line 77: Why were BMIM ILs chosen for this work? Why not EMIM? They tend to generally have lower toxicity?

Line 95:  How stable are the ILs at a temperature of 150°C, especially for longer amounts of  time? Have you analysed them to verify, that they are not at least partially decomposed?

Line 153: ibid. The decomposition is probably not only in the polyphenols, but also in the IL as well. Please verify.

Author Response

The authors are grateful to the reviewer for valuable comments and remarks, which allowed to improve the quality of the article. The authors' answers to the questions and comments are given below.

Line 77: Why were BMIM ILs chosen for this work? Why not EMIM? They tend to generally have lower toxicity?

Indeed, bmim cation-based ionic liquids can exhibit greater toxicity compared to emim cation-based ionic liquids. However, the aim of this study was to demonstrate the possibility of using bmim-based ILs to dissolve algal biomass with the subsequent isolation of polysaccharide and polyphenolic fractions according to the methodology developed earlier for wood biomass (Line 302-313). Emim-based ILs possess similar properties and can be used instead of bmim-based ones.

Line 95:  How stable are the ILs at a temperature of 150°C, especially for longer amounts of time? Have you analysed them to verify, that they are not at least partially decomposed?

Line 153: ibid. The decomposition is probably not only in the polyphenols, but also in the IL as well. Please verify.

Indeed, not all ionic liquids used in our study demonstrate high stability under continuous heating at temperatures above 120-150°C. Previously obtained data indicate noticeable degradation of [bmim]OAc under the above conditions. In this regard, reasoning has been added to the manuscript (Lines 120-124): “Considering the previous data on the thermostability of the ionic liquids used [20], prolonged heating of the reaction mixture at 120-150 °C cannot be recommended for [bmim]OAc which is prone to partial decomposition under these conditions. However, samples isolated with [bmim]OAc at 120-150 °C were further used for comparison purposes.” It should also be mentioned that the main purpose of the work was not to select optimal conditions, but to demonstrate the possibility of using dialkylimidazolium ionic liquids for fractionation of brown algal biomass.

Reviewer 2 Report

Comments and Suggestions for Authors

This paper presents three different 1-Butyl-3-methylimidazolium-based ionic liquids that were used to dissolve brown algae biomass with selective antisolvent precipitation to separate polysaccharide and polyphenolic constituents. This interesting study requires several improvements before being published in Molecules.

- Pyridinium salts have been used in other studies, the authors should discuss the influence of the cation, these salts present good thermal and chemical stability compatible with the protocol of this study. Thus, the chemical interactions of the cation with the possible participation of a carbene intermediate can be discussed to facilitate the dissolution of biopolymers.

- At 80 °C, it is specified that the greater dissolution of algae biomass (30%) in [bmim]OAc and [bmim]MeSO4 can be explained by a cleavage of the ether and ester bonds in the biopolymers. What is the evidence for this hypothesis?

- For the selective isolation of polysaccharide and polyphenol components, the authors use sequential precipitation with acetone and water. Other antisolvents have been used to study a modification of the composition and improve the yields (F1)?

- From 80 to 150 °C, the quantity of isolated polysaccharides increases but it does not correspond to the percentage of dissolved biomass. The authors propose hydrolysis under the action of an IL-water mixture with [bmim]MeSO4 and [bmim]Cl (acidic anions promoting hydrolysis). This result requires the study of LIs that are more hydrophobic (BF4 or PF6 anion) and less favorable to hydrogen bonds (pyridinum salts).

- The chemical composition of fraction F2 could present several amino acids, a more complete analysis by mass spectrometry could make it possible to identify these compounds.

Based on additional information on all of these points, this work can be published in Molecules.

Comments on the Quality of English Language

Minor editing of English language required

Author Response

The authors are grateful to the reviewer for valuable comments and remarks, which allowed to improve the quality of the article. The authors' answers to the questions and comments are given below.

- Pyridinium salts have been used in other studies, the authors should discuss the influence of the cation, these salts present good thermal and chemical stability compatible with the protocol of this study. Thus, the chemical interactions of the cation with the possible participation of a carbene intermediate can be discussed to facilitate the dissolution of biopolymers.

Indeed, pyridinium salts can exhibit good thermal and chemical stability. However, most of the works using alkylimidazolium and pyridinium ILs are aimed at isolation of polysaccharides for subsequent hydrolysis and lipid component of algae. This makes it difficult to estimate the effect of the cation on the solubility of algal biomass based on literature data (Line 68-75). However, the aim of our work was to evaluate the possibility of using dialkylimidazolium ionic liquids for the dissolution and subsequent fractionation of brown algal biomass since this type of ionic liquids proved itself well earlier in fractionation of wood biomass. The study of the IL cation effect may be the aim of the future works.

- At 80 °C, it is specified that the greater dissolution of algae biomass (30%) in [bmim]OAc and [bmim]MeSO4 can be explained by a cleavage of the ether and ester bonds in the biopolymers. What is the evidence for this hypothesis?

This assumption is based on the known ability of these ILs to cleavage the ether and ester bonds in the biopolymers during their dissolution. The corresponding sentence with the appropriate references has been added to the text (Line 64-68).

- For the selective isolation of polysaccharide and polyphenol components, the authors use sequential precipitation with acetone and water. Other antisolvents have been used to study a modification of the composition and improve the yields (F1)?

The authors did not use any other antisolvents in this study. The aim of this study was to demonstrate the feasibility of the previously proposed approach for brown algal biomass fractionation involving the precipitation of biopolymers with acetone and water (Line 304-315).

- From 80 to 150 °C, the quantity of isolated polysaccharides increases but it does not correspond to the percentage of dissolved biomass. The authors propose hydrolysis under the action of an IL-water mixture with [bmim]MeSO4 and [bmim]Cl (acidic anions promoting hydrolysis). This result requires the study of LIs that are more hydrophobic (BF4 or PF6 anion) and less favorable to hydrogen bonds (pyridinum salts).

Indeed, the use of other cations and anions will allow to draw more detailed conclusions about the processes occurring during dissolution of biopolymers in ILs. However, this was beyond the scope of the presented work, the purpose of which was to demonstrate the possibility of using dialkylimidazolium ionic liquids for fractionation of brown algal biomass.

- The chemical composition of fraction F2 could present several amino acids, a more complete analysis by mass spectrometry could make it possible to identify these compounds.

The main compositions of the isolated compounds were characterized by IR and NMR spectroscopy techniques. Application of mass spectrometry methods and more detailed analysis of the isolated components will be presented in ongoing paper.

Round 2

Reviewer 2 Report

Comments and Suggestions for Authors

A response was given by the authors on the various points. Overall, clarification has been provided regarding this research work. However, it is necessary to specify in the manuscript that the precise characterization of the amino acids potentially present in the F2 fraction is not possible here and that additional work is currently in progress on the characterization of these amino acids.

Comments on the Quality of English Language

Minor editing of English language required

Author Response

The authors are grateful to the reviewer for the valuable comment. Following the suggestion of an academic editor, the additional sentence has been added to the manuscript (Lines 288-290): "However, identification of all amino acids present in the fraction F2 is currently impossible. Further research is underway".